# Reading Wikipedia to Answer Open-Domain Questions

## Abstract

This paper proposes to tackle open-domain question answering using Wikipedia as the unique knowledge source: the answer to any factoid question is a text span in a Wikipedia article. This problem combines the challenges of document retrieval, to find relevant articles, and of machine comprehension of text to identify the answer spans from those articles. Our approach combines a search component based on bigram hashing and TF-IDF matching with a multi-layer recurrent neural network model trained to detect answers in Wikipedia paragraphs. Our experiments on multiple existing QA datasets indicate that (1) both modules are highly competitive with respect to existing counterparts and (2) multitask learning using distant supervision on their combination is an effective complete system on this challenging task.

## 1 Introduction

This paper considers the problem of answering factoid questions in an open-domain setting using Wikipedia as the unique knowledge source, as one does when looking for answers in an encyclopedia. Wikipedia is a constantly evolving source of detailed information that could facilitate intelligent machines — if they are able to leverage its power. Unlike knowledge bases (KBs) such as Freebase (Bollacker et al., 2008) or DB-Pedia (Auer et al., 2007), which are easier for computers to process but too sparsely populated for open-domain question answering (Miller et al., 2016), Wikipedia contains up-to-date knowledge that humans are interested in, but is designed for humans, not machines, to read.

Using Wikipedia articles as the knowledge source causes the task of question answering (QA) to combine the challenges of both large-scale open-domain QA and of machine comprehension of text. In order to answer any question, one must first retrieve the few relevant articles among more than 5 millions items, and scan them carefully to identify the answer. Our work treats Wikipedia as a collection of articles and does not rely on its internal graph structure. As a result, our approach is generic and could be switched to another collection of documents.

Large-scale QA systems like IBM's DeepQA (Ferrucci et al., 2010) rely on multiple sources to answer: Wikipedia can be one of them but it is also paired with KBs, dictionaries, and even news articles, books, etc. As a result, such systems heavily rely on information redundancy among the sources to answer correctly. Having a single knowledge source forces the model to be very precise while searching for an answer as the evidence might appear only once. This challenge thus encourages research in the ability of a machine to read, a key motivation for the machine comprehension subfield and the creation of datasets such as SQuAD (Rajpurkar et al., 2016), CNN/Daily Mail (Hermann et al., 2015) and CBT (Hill et al., 2016).

However, those machine comprehension resources typically assume that a short piece of relevant text is already identified and given to the model, which is not realistic for building an open-domain QA system. In sharp contrast, methods that use KBs or information retrieval over documents have to employ search as an integral part of the solution. Instead we aim at the setting of simultaneously maintaining the challenge of machine comprehension, which requires the deep understanding of text, while keeping the realistic constraint of searching over a large open resource.

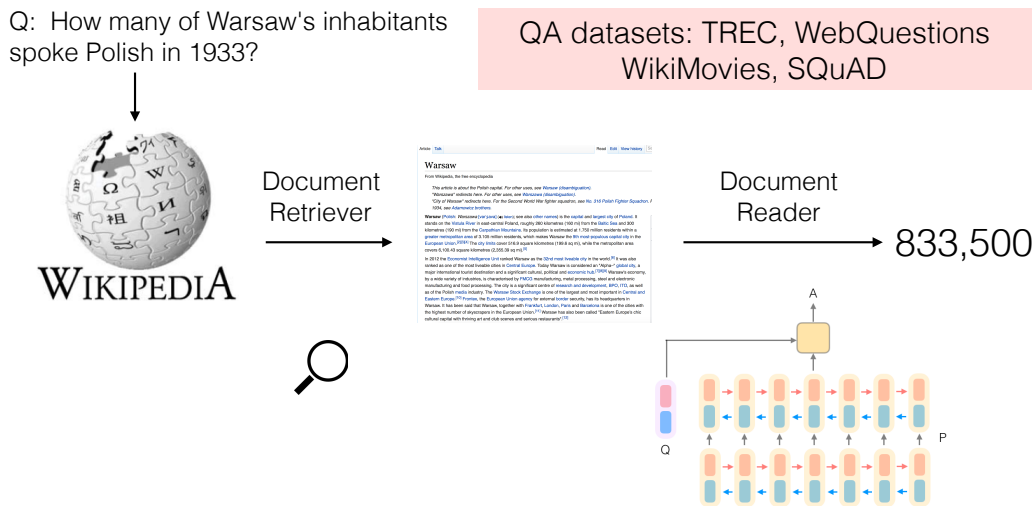

Figure 1: An overview of our question answering system DrWiki.

In this paper, we show how multiple existing QA datasets can be used to evaluate this framework by requiring an open-domain system to perform well on all of them at once. We develop DrWiki, a strong system for question answering from Wikipedia composed of: (1) Document Retriever, a module using bigram hashing and TF-IDF matching designed to, given a question, efficiently return a subset of relevant articles and (2) Document Reader, a multi-layer recurrent neural network machine comprehension model trained to detect answer spans in those few returned documents. Figure 1 gives an illustration of DrWiki.

Our experiments show that Document Retriever outperforms the built-in Wikipedia search engine and that Document Reader achieves state-of-the-art results on the very competitive SQuAD benchmark (Rajpurkar et al., 2016). Finally, our full system is evaluated using multiple benchmarks. In particular, we show that performance is improved across almost all datasets through the use of multi-task learning and distant supervision compared to single task training.

## 2   Related Work

Open-domain QA was originally defined as finding answers in collections of unstructured documents, following the setting of the annual TREC competitions[1]. With the development of KBs, many recent innovations have occurred in the context of QA from KBs with the creation of re-

sources like WebQuestions (Berant et al., 2013) and SimpleQuestions (Bordes et al., 2015) based on the Freebase KB (Bollacker et al., 2008), or on automatically extracted KBs, e.g., OpenIE triples and NELL (Fader et al., 2014). However, KBs have inherent limitations (incompleteness, fixed schemas) that motivated researchers to return to the original setting of answering from raw text.

A second motivation to cast a fresh look at this problem is that of machine comprehension of text, i.e., answering questions after reading a short text or story. That subfield has made considerable progress recently thanks to new deep learning architectures like attention-based and memory-augmented neural networks (Bahdanau et al., 2015; Weston et al., 2015; Graves et al., 2014) and the release of new training and evaluation datasets like QuizBowl (Iyyer et al., 2014), CNN/Daily Mail based on news articles (Hermann et al., 2015), CBT based on children books (Hill et al., 2016), or SQuAD (Rajpurkar et al., 2016) and WikiReading (Hewlett et al., 2016), both based on Wikipedia. An objective of this paper is to test how such new methods can perform in an open-domain QA framework.

QA using Wikipedia as a resource has been explored previously. The authors of (Ryu et al., 2014) perform open-domain QA using a Wikipedia-based knowledge model. They combine article content with multiple other answer matching modules based on different types of semi-structured knowledge such as infoboxes, article structure, category structure, and defini-

---

[1] http://trec.nist.gov/data/qamain.html

tions. Similarly (Ahn et al., 2004) also combine Wikipedia as a text resource with other resources, in this case with information retrieval over other documents. Buscaldi and Rosso (2006) also mine knowledge from Wikipedia for QA. Instead of using it as a resource for seeking the answers to questions, they focused on validation of the answers as returned by their QA system, and the use of Wikipedia categories for determining a set of patterns that should fit with the expected answer. In our work, we consider the comprehension of text only, and use the Wikipedia text documents as a sole resource for the reasons described in the introduction, that is to help focus on the machine comprehension task.

There are a number of highly developed full pipeline QA approaches using Wikipedia as a resource, including Microsoft's AskMSR (Brill et al., 2002), IBM's DeepQA (Ferrucci et al., 2010) and YodaQA (Baudiš, 2015; Baudiš and Šedivỳ, 2015), the latter of which is open source and hence reproducible for comparison purposes. AskMSR is a search-engine based QA system that relies on "data redundancy rather than sophisticated linguistic analyses of either questions or candidate answers", i.e., it does not focus on machine comprehension, as we do. DeepQA is a very sophisticated system that relies on both unstructured information including text documents as well as structured data such as KBs, databases and ontologies to generate candidate answers or vote over evidence. YodaQA is an open source system modeled after DeepQA, similarly combining websites, information extraction, databases and Wikipedia in particular. While our comprehension task is far more challenging as we do not allow the leveraging of extra resources, comparing to these methods as an "upper bound" benchmark of what we could achieve is a useful datapoint.

Multitask learning (Caruana, 1998) and task transfer (e.g., in the computer vision community using ImageNet (Huh et al., 2016)) have a rich history in machine learning, as well as in NLP in particular (Collobert and Weston, 2008). Several works have attempted to combine multiple QA training datasets via multitask learning to (i) achieve improvement across the datasets via task transfer; and (ii) to provide a single general system capable of asking different kinds of questions due to the inevitably different data distributions across the source datasets. Fader et al.

(2014) used WebQuestions, TREC and WikiAnswers with four KBs as knowledge sources and reported improvement on the latter two datasets through multitasking. Bordes et al. (2015) combined WebQuestions and SimpleQuestions using distant supervision with Freebase as the KB to give slight improvements on both datasets, although poor performance was reported training on only one dataset and testing on the other, showing that task transfer is indeed a challenging subject; see also (Kadlec et al., 2016) for a similar conclusion. Our work follows similar themes, but in the setting of having to retrieve and then read text documents, rather than using a KB.

## 3 Our System: DrWiki

In the following we describe our system DrWiki for open-domain Wikipedia question answering which consists of two components: (1) the Document Retriever module for finding relevant articles and (2) a machine comprehension model, Document Reader, for extracting answers from a single document or a small collection of documents.

### 3.1 Document Retriever

Following classical QA systems, we use an efficient (non-machine learning) document retrieval system to first narrow our search space and focus on reading only articles that are likely to be relevant. A simple inverted index lookup followed by term vector model scoring performs quite well on this task for many question types, compared to the built-in ElasticSearch based Wikipedia Search API (Gormley and Tong, 2015). Articles and questions are compared as TF-IDF weighted bag-of-word vectors. We further improve our system by taking local word order into account with n-gram features. Our best performing system uses bigram counts while preserving speed and memory efficiency by using the hashing of (Weinberger et al., 2009) to map the bigrams to $2^{24}$ bins with an unsigned *murmur3* hash.

We use Document Retriever as the first part of our full model, by setting it to return 5 Wikipedia articles given any question. Those articles are then processed by Document Reader.

### 3.2 Document Reader

Our Document Reader model is inspired by the recent success of neural network models on machine comprehension tasks, in a similar spirit to the *At-*

*tentiveReader* described in (Hermann et al., 2015; Chen et al., 2016).

Given a question $q$ consisting of $l$ tokens $\{q_1, \ldots, q_l\}$ and a document or a small set of documents of $n$ paragraphs where paragraph $p^{(k)}$ consists of $l_k$ tokens $\{p_1^{(k)}, \ldots, p_{l_k}^{(k)}\}$ for $1 \leq k \leq n$, we develop an RNN model that we apply to each paragraph in turn and then finally aggregate the predicted answers. Our method works as follows:

**Paragraph encoding**   We first represent each token $p_i$ in a paragraph $p$ as a feature vector $\tilde{\mathbf{p}}_i \in \mathbb{R}^d$ and pass it as the input to a multi-layer recurrent neural network and thus obtain:

$$\{\mathbf{p}_1, \ldots, \mathbf{p}_m\} = RNN(\{\tilde{\mathbf{p}}_1, \ldots, \tilde{\mathbf{p}}_m\}),$$

where $\mathbf{p}_i$ is expected to encode useful context information around token $p_i$. Specifically, we choose to use a three-layer bidirectional long short-term memory network (LSTM) with $h$ hidden units at each layer/direction as our RNN architecture and concatenate all the hidden units in the end. Therefore the resulting $\mathbf{p}_i$ contains $6h$ dimensions.

The feature vector $\tilde{\mathbf{p}}_i$ is comprised of the following parts:

- *Word embeddings*: $f_{emb}(p_i) = \mathbf{E}(p_i)$. We use the 300-dimensional Glove word embeddings trained from 840B Web crawl data (Pennington et al., 2014). We keep most of the pre-trained word embeddings fixed and only fine-tune the $1{,}000$ most frequent words because we think that some question words such as *what*, *how*, *which*, *many* could be highly important for QA systems.

- *Exact match*: $f_{exact\_match}(p_i) = \mathbb{I}(p_i \in q)$. We use three simple binary features, indicating whether $p_i$ can be exactly matched to one question word in $q$, either in its original, lowercased or lemma form. These simple features turn out to be extremely helpful, as we will show in Section 5.

- *Token features*: $f_{token}(p_i) = (\text{POS}(p_i), \text{NER}(p_i), \text{TF}(p_i))$. We also add a few manual features which reflect some properties of token $p_i$ in its context, which include its part-of-speech (POS) and named entity recognition (NER) tags and its (normalized) term frequency (TF).

- *Aligned question embedding*: following (Lee et al., 2016) and other recent works, the last part we incorporate is an aligned question embedding $f_{align}(p_i) = \sum_j \alpha(\mathbf{E}(p_i), \mathbf{E}(q_j))\mathbf{E}(q_j)$, where $\alpha_j \propto \frac{\mathbf{E}(p_i)^{\intercal}\mathbf{E}(q_j)}{\sum_{j'}\mathbf{E}(p_i)^{\intercal}\mathbf{E}(q_{j'})}$, which captures the similarity between $p_i$ and all the question words in $q$. Compared to the *exact match* features, these features add soft alignments between similar but non-identical words (e.g., *car* and *vehicle*).

**Question encoding**   The question encoding is simpler, as we only apply another *RNN* on top of the word embeddings of $q_i$ and combine the resulting hidden units into one single vector: $\{\mathbf{q}_1, \ldots, \mathbf{q}_l\} \rightarrow \mathbf{q}$. We compute $\mathbf{q} = \sum \beta(\mathbf{q}_i)\mathbf{q}_i$ where $\beta(\cdot)$ is a normalized weighting function which learns the importance of each question word.

**Prediction**   At the paragraph level, the goal is to predict the span of tokens that is most likely the correct answer. We take the the paragraph vectors $\{\mathbf{p}_1, \ldots, \mathbf{p}_m\}$ and the question vector $\mathbf{q}$ as input, and simply train two classifers independently for predicting the two ends of the span. Concretely, we use a bilinear term to capture the similarity between $\mathbf{p}_i$ and $\mathbf{q}$:

$$
\begin{aligned}
P_{start}(i) &= softmax\left(\mathbf{p}_i W_s \mathbf{q}\right) \\
P_{end}(i) &= softmax\left(\mathbf{p}_i W_e \mathbf{q}\right)
\end{aligned}
$$

During prediction, we choose the best span from token $i$ to token $i'$ such that $i \leq i' \leq i + 15$ and $P_{start}(i) \times P_{end}(i')$ is maximized. To make scores compatible across paragraphs in one or several retrieved documents, we replace the softmax with an unnormalized exponential. Our final prediction is then the argmax over all considered paragraph spans.

## 4   Data

Our work relies on three types of data: (1) Wikipedia that serves as our knowledge source for finding answers, (2) the SQuAD dataset which is our main resource to train Document Reader and (3) three more QA datasets (CuratedTREC, WebQuestions and WikiMovies) that in addition to SQuAD, are used to test the open-domain QA abilities of our full system, and to evaluate the ability of our model to learn from multitasking and

| Dataset | Example | Article / Paragraph |
|---|---|---|
| SQuAD | **Q**: How many provinces did the Ottoman empire contain in the 17th century? <br> **A**: 32 | **Article**: Ottoman Empire <br> **Paragraph**: ... At the beginning of the 17th century the empire contained 32 provinces and numerous vassal states. Some of these were later absorbed into the Ottoman Empire, while others were granted various types of autonomy during the course of centuries. |
| CuratedTREC | **Q**: What U.S. state's motto is "Live free or Die"? <br> **A**: New Hampshire | **Article**: Live Free or Die <br> **Paragraph**: "Live Free or Die" is the official motto of the U.S. state of New Hampshire, adopted by the state in 1945. It is possibly the best-known of all state mottos, partly because it conveys an assertive independence historically found in American political philosophy and partly because of its contrast to the milder sentiments found in other state mottos. |
| WebQuestions | **Q**: What part of the atom did Chadwick discover?[†] <br> **A**: neutron | **Article**: Atom <br> **Paragraph**: ... The atomic mass of these isotopes varied by integer amounts, called the whole number rule. The explanation for these different isotopes awaited the discovery of the neutron, an uncharged particle with a mass similar to the proton, by the physicist James Chadwick in 1932. ... |
| WikiMovies | **Q**: Who wrote the film Gigli? <br> **A**: Martin Brest | **Article**: Gigli <br> **Paragraph**: Gigli is a 2003 American romantic comedy film written and directed by Martin Brest and starring Ben Affleck, Jennifer Lopez, Justin Bartha, Al Pacino, Christopher Walken, and Lainie Kazan. |

Table 1: Example training data from each QA dataset. In each case we show an associated article where distant supervision (DS) correctly identified the answer within it, which is highlighted (See Sec. 4.4).

| Dataset | Train | | Test |
|---|---|---|---|
| | Plain | DS | |
| SQuAD | 87,385 | 31,775 | 10,570[†] |
| CuratedTREC | 1,486* | 3,464 | 694 |
| WebQuestions | 3,778* | 4,602 | 2,032 |
| WikiMovies | 96,185* | 36,301 | 9,952 |

Table 2: Number of questions for each dataset used in this paper. DS: distantly supervised training data. *: These training sets are not used as is because no paragraph is associated with each question. [†]: Corresponds to SQuAD development set.

distant supervision. Statistics of the datasets are given in Table 2.

### 4.1 Wikipedia (Knowledge Source)

We use the 2016-12-21 dump[2] of English Wikipedia for all of our full-scale experiments as the knowledge source used to answer questions. For each page, only the plain text is extracted and all structured data sections such as lists and figures are stripped[3]. After discarding disambiguation pages, we retain 5,236,178 articles consisting

---

[2] https://dumps.wikimedia.org/enwiki/latest

[3] https://github.com/attardi/wikiextractor

of 9,264,930 unique uncased token types.

### 4.2 SQuAD

The Stanford Question Answering Dataset (SQuAD) (Rajpurkar et al., 2016) is a dataset for machine comprehension based on Wikipedia. The dataset contains 90k/10k train/development examples with a large hidden test set, that can only be accessed by the SQuAD creators. Each example is composed of a paragraph extracted from a Wikipedia article and an associated human-generated question. The answer is always a span from this paragraph and a model is given credit if its predicted answer matches it. Two evaluation metrics are used: Exact string match (EM) and F1 score, which measures the weighted average of precision and recall at the token level.

In the following, we use SQuAD for training and evaluating our Document Reader for the standard machine comprehension task given the relevant paragraph as defined in (Rajpurkar et al., 2016). For the task of evaluating open-domain question answering over Wikipedia, we use the SQuAD development set QA pairs only, and we ask systems to uncover the correct answer spans *without* having access to the associated paragraphs. That is, a model is required to answer a given question given the whole of Wikipedia as a resource, it is *not* given the relevant paragraph as

| Dataset | Wiki Search | Doc. Retriever | |
|---|---|---|---|
| | | plain | +bigrams |
| SQuAD | 63.5 | 76.1 | **77.8** |
| CuratedTREC | 82.9 | 85.3 | **86.2** |
| WebQuestions | 74.1 | **75.8** | 74.6 |
| WikiMovies | 61.0 | 53.3 | **68.7** |

Table 3: Document retrieval results. % of questions for which the answer segment appears in one of the top 5 pages returned by the method.

in the standard SQuAD setting.

### 4.3 Open-domain QA Evaluation Resources

SQuAD is one of the largest general purpose QA datasets currently available. SQuAD questions have been collected via a process involving showing a paragraph to each human annotator and asking them to write a question. As a result, their distribution is quite specific. We hence propose to train and evaluate our system on other datasets developed for open-domain QA that have been constructed in different ways (not necessarily in the context of answering from Wikipedia).

**CuratedTREC** This dataset is based on the benchmarks from the TREC QA tasks that have been curated by Baudiš and Šedivỳ (2015). We use the large version, which contains a total of 2,180 questions extracted from the datasets from TREC 1999, 2000, 2001 and 2002.[4]

**WebQuestions** Introduced in (Berant et al., 2013), this dataset is built to answer questions from the Freebase KB. It was created by crawling questions through the Google Suggest API, and then obtaining answers using Amazon Mechanical Turk. We converted each answer to text by using entity names so that the dataset does not reference Freebase IDs and is purely made of plain text question-answer pairs.

**WikiMovies** This dataset, introduced in (Miller et al., 2016), contains 96k question-answer pairs in the domain of movies. Originally created from Freebase, the examples are built such that they can also be answered by using a subset of Wikipedia as the knowledge source (the title and the first section of articles from the movie domain).

---

[4]This dataset is available at `https://github.com/brmson/dataset-factoid-curated`.

### 4.4 Distantly Supervised Data

All the QA datasets presented above contain training portions, but CuratedTREC, WebQuestions and WikiMovies only contain question-answer pairs, and not an associated document or paragraph as in SQuAD, and hence cannot be used for training Document Reader directly. Following previous work on distant supervision (DS) for relation extraction (Mintz et al., 2009), we use a procedure to automatically associate paragraphs to such training examples, and then add these examples to our training set.

We use the following process for each question-answer pair to build our training set. First, we run Document Retriever on the question to retrieve the top 5 Wikipedia articles. All paragraphs from those articles without an exact match of the known answer are directly discarded. All paragraphs shorter than 25 or longer than 1500 characters are also filtered out. If any named entities are detected in the question, we remove any paragraph that does not contain them at all. For every remaining paragraph in each retrieved page, we score all positions that match an answer using unigram and bigram overlap between the question and a 20 token window, and we keep the paragraph with the highest overlap. If there is no paragraph with non-zero overlap, the example is discarded; otherwise we add it to our DS training dataset.

Examples of generated examples are given in Table 1 and the size of each DS dataset is given in the third column (DS) of Table 2.

Note that we can also generate additional DS data for SQuAD by trying to find mentions of the answers in other paragraphs other than the original paragraph provided in the dataset (from other pages or the same page that the given paragraph was in). As shown in Table 2, we can generate such additional examples for around a third of the original SQuAD training examples.

## 5 Experiments

This section first presents evaluations of our Document Retriever and Document Reader modules separately, and then describes tests of their combination, DrWiki, for open-domain QA on the full Wikipedia.

### 5.1 Finding Relevant Articles

We first examine the performance of our Document Retriever module on all the QA datasets. Ta-

| Method | Dev | | Test | |
|---|---|---|---|---|
| | EM | F1 | EM | F1 |
| Dynamic Coattention Networks (Xiong et al., 2016) | 65.4 | 75.6 | 66.2 | 75.9 |
| BiDAF (Seo et al., 2016) | 67.7 | 77.3 | 68.0 | 77.3 |
| Multi-Perspective Matching (Wang et al., 2016)[†] | 66.1 | 75.8 | 68.9 | 77.8 |
| R-net[‡] | n/a | n/a | 71.3 | 79.7 |
| DrWiki (Our model, Document Reader Only) | **69.5** | **78.8** | 70.0 | 79.0 |

Table 4: Evaluation results on the SQuAD dataset (single model only). All the test results reflect the SQuAD leaderboard[6] as of Feb 6, 2017. [†]: Test set results have been updated on the leaderboard after the paper submission. [‡]: Paper unavailable.

ble 3 compares the performance of the two approaches described in Section 3.1 with that of the Wikipedia Search Engine[7] for the task of finding articles that contain the answer given a question. Specifically, we compute the ratio of questions for which the text span of any of their associated answers appear in at least one the top 5 relevant pages returned by each system. Results on all datasets indicate that our simple approach outperforms Wikipedia Search, especially with bigram hashing.

## 5.2 Reader Evaluation on SQuAD

Next we evaluate our Document Reader component on the standard SQuAD evaluation (Rajpurkar et al., 2016).

**Implementation details** We use a 3-layer bidirectional LSTMs with $h = 128$ hidden units for both paragraph and question encoding. We apply the Stanford CoreNLP toolkit (Manning et al., 2014) for tokenization and also generating lemma, part-of-speech, and named entity manual features.

Lastly, all the training examples are sorted by the length of paragraph and divided into minibatches of 32 examples each. We use *Adamax* for optimization as described in (Kingma and Ba, 2014). Dropout with $p = 0.3$ is applied to word embeddings and all the hidden units of LSTMs.

**Result and analysis** Table 4 presents our evaluation results on both development and test sets. SQuAD has been a very competitive machine comprehension benchmark since its creation and we only list the best-performing systems in the table. Our system (single model) can achieve 70.0% exact match and 79.0% F1 scores on the test set, which surpasses all the published results and can

[7]We used the Wikipedia Search API https://www.mediawiki.org/wiki/API:Search.

| Features | F1 |
|---|---|
| Full | 78.4 |
| No $f_{token}$ | 77.9 (-0.5) |
| No $f_{exact\_match}$ | 77.2 (-1.2) |
| No $f_{aligned}$ | 76.1 (-2.3) |
| No $f_{aligned}$ and $f_{exact\_match}$ | 59.4 (-19.0) |

Table 5: Feature ablation analysis of the paragraph representations of our Document Reader. Results are reported on the SQuAD development set, using a slightly inferior model compared to Table 4.

match the top performance on the SQuAD leaderboard at the time of writing. Additionally, we think that our model is conceptually simpler than most of the existing systems. We conducted an ablation analysis on the feature vector of paragraph tokens. As shown in Table 5 all the features contribute to the performance of our final system. Without the aligned question embedding feature (only word embedding and a few manual features), our system is still able to achieve F1 over 76%. More interestingly, if we remove both $f_{aligned}$ and $f_{exact\_match}$, the performance drops dramatically, so we conclude that both features play a similar but complementary role in the feature representation related to the paraphrased nature of a question vs. the context around an answer.

## 5.3 Full Wikipedia Question Answering

Finally, we assess the performance of our full system DrWiki for answering open-domain questions using the four datasets introduced in Section 4. We compare three versions of DrWiki which evaluate the impact of using distant supervision and multitask learning across the training sources provided to Document Reader (Document Retriever remains the same for each case):

| Dataset | YodaQA | DrWiki | | |
|---|---|---|---|---|
| | | SQuAD | +Fine-tune (DS) | +Multitask (DS) |
| SQuAD *(All Wikipedia)* | n/a | 26.7 | 29.2 | 29.6 |
| CuratedTREC | 31.3 | 19.7 | 25.5 | 24.5 |
| WebQuestions | 39.8 | 19.6 | 19.3 | 18.8 |
| WikiMovies | n/a | 23.6 | 33.1 | 34.4 |

Table 6: Full Wikipedia results. Top-1 accuracy (in %). +Fine-tune (DS): Document Reader models trained on SQuAD and fine-tuned on each DS training set independently. +Multitask (DS): Document Reader single model trained on SQuAD and all the distant supervision (DS) training sets jointly. YodaQA results are extracted from `https://github.com/brmson/yodaqa/wiki/Benchmarks` and use additional resources other than Wikipedia such as Freebase and DBpedia, see Section 2.

- SQuAD setting: A single Document Reader model is trained on the SQuAD training set only and used on all evaluation sets.

- Fine-tune (DS): A Document Reader model is pre-trained on SQuAD and then fine-tuned for each dataset independently using its dedicated distant supervision (DS) training set.

- Multitask (DS) setting: A single Document Reader model is jointly trained on the SQuAD training set and *all* the DS sources.

**Results** Table 6 presents the results. Despite the difficulty of the task compared to machine comprehension (where you are given the right paragraph) and unconstrained QA (using redundant resources), DrWiki still provides reasonable performance across all four datasets.

We are interested in a single full system that can answer any question using Wikipedia. The single model trained only on SQuAD is outperformed on three of the datasets by the multitask system that uses distant supervision. However performance when training on SQuAD alone is not far behind indicating that task transfer is occurring. The majority of the improvement from SQuAD to Multiclass(DS) however is likely not from task transfer as fine-tuning on each dataset alone using DS also gives improvements, showing that it is the introduction of extra data in the same domain that helps. Nevertheless, the best single model that we can find is our overall goal, and that is the Multitask(DS) system.

We compare to an unconstrained QA system using redundant resources (not just Wikipedia), YodaQA (Baudiš, 2015), giving results which were previously reported on CuratedTREC and WebQuestions. Despite the increased difficulty of our task it is reassuring that our performance is not too far behind on CuratedTREC (31.3 vs. 25.5). The gap is slightly bigger on WebQuestions likely because that is based on Freebase which YodaQA uses directly.

DrWiki's performance on SQuAD compared to its Document Reader component on machine comprehension in Table 4 shows a large drop (from 69.5 to 26.7) as we now are given Wikipedia to read, not a single paragraph. Given the correct document (but not the paragraph) we can achieve 49.6 indicating many false positives come from highly topical sentences. This is despite the fact that the Document Retriever works relatively well (77.8% of the time retrieving the answer, see Table 3). Overall, our results indicate that we have identified a key challenging task for researchers to focus on.

# 6 Conclusion

We studied the task of open-domain QA using Wikipedia as the unique knowledge source. This brings unique challenges for machine comprehension systems where integrating search, distant supervision and multitask learning provides an effective complete system, whereas machine comprehension systems alone cannot solve the overall task. Evaluating the individual components and the overall system across multiple benchmarks showed the efficacy of our approach.

Future work should aim to improve over our DrWiki system. Two obvious angles of attack are: (i) incorporate the fact that Document Reader aggregates over multiple paragraphs and documents directly in the training, as it currently trains on paragraphs independently; and (ii) perform end-to-end training across the Document Retriever and Document Reader pipeline, rather than independent systems.

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
