# Peer review of "Reading Wikipedia to Answer Open-Domain Questions"

_ACL 2017 — decision unknown_

[Official Review · Reviewer 1 · rating 3 · confidence 4]
soundness 3 · originality 4 · clarity 5 · impact 5 · substance 3 · appropriateness 5 · meaningful comparison 5 · presentation format Poster

- Strengths:
*- Task
*- Simple model, yet the best results on SQuAD (single model0
*- Evaluation and comparison

- Weaknesses:
*- Analysis of errors/results (See detailed comments below)

- General Discussion:
In this paper the authors present a method for directly querying Wikipedia to
answer open domain questions. The system consist of two components - a module
to query/fetch wikipedia articles and a module to answer the question given the
fetched set of wikipedia articles. 

The document retrieval system is a traditional IR system relying on term
frequency models and ngram counts.  The answering system uses a feature
representation for paragraphs that consists of word embeddings, indicator
features to determine whether a paragraph word occurs in a question,
token-level features including POS, NER etc and a soft feature for capturing
similarity between question and paragraph tokens in embedding space. A combined
feature representation is used as an input to a bi-direction LSTM RNN for
encoding. For questions an RNN that works on the word embeddings is used. 
These are then used to train an overall classifier independently for start and
end spans of sentences within a paragraph to answer questions.

The system has been trained using different Open Domain QA datasets such as
SQuAD and WebQuestions by modifying the training data to include articles
fetched by the IR engine instead of just the actual correct document/passage.

Overall, an easy to follow interesting paper but I had a few questions:
1) The IR system has a Accuracy@5 of over 75 %, and individually the document
reader performs well and can beat the best single models on SquAD. What
explains the significant drop in Table 6. The authors mention that instead of
the fetched results, if they test using the best paragraph the accuracy reaches
just 0.49 (from 0.26) but that is still significantly below the 0.78-79 in the
SQuAD task.  So, presumably the error is this large because the neural network
for matching isnt doing as good a job in learning the answers when using the
modified training set (which includes fetched articles) instead of the case
when training and testing is done for the document understanding task. Some
analysis of whats going on here should be provided. What was the training
accuracy in the both cases? What can be done to improve it? To be fair, the
authors to allude to this in the conclusion but I think it still needs to be
part of the paper to provide some meaningful insights.

2) I understand the authors were interested in treating this as a pure machine
comprehension task and therefore did not want to rely on external sources such
as Freebase which could have helped with entity typing        but that would have
been interesting to use. Tying back to my first question -- if the error is due
to highly relevant topical sentences as the authors mention, could entity
typing have helped?

The authors should also refer to QuASE (Sun et. al 2015 at WWW2015) and similar
systems in their related work. QuASE is also an Open domain QA system that
answers using fetched passages - but it relies on the web instead of just
Wikipedia.

[Official Review · Reviewer 2 · rating 3 · confidence 3]
soundness 3 · originality 4 · clarity 4 · impact 5 · substance 3 · appropriateness 5 · meaningful comparison 5 · presentation format Oral Presentation

- Strengths:

The authors focus on a very challenging task of answering open-domain question
from Wikipedia. Authors have developed 1) a document retriever to retrieve
relevant Wikipedia articles for a question, and 2) Document retriever to
retrieve the exact answer from the retrieved paragraphs. 
Authors used Distant Supervision to fine-tune their model. Experiments show
that the document reader performs better than WikiSearch API, and Document
Reader model does better than some recent models for QA.

- Weaknesses:
The final results are inferior to some other models, as presented by the
authors. Also, no error analysis is provided.

- General Discussion:

The proposed systems by the authors is end-to-end and interesting. However, I
have some concerns below.

Document Retriever: Authors have shown a better retrieval performance than Wiki
Search. However, it is not described as to how exactly the API is used.
WikiSearch may not be a good baseline for querying "questions" (API suits
structured retrieval more). Why don't the authors use some standard IR
baselines for this?

Distant Supervision: How effective and reliable was distant supervision?
Clearly, the authors had to avoid using many training examples because of this,
but whatever examples the authors could use, what fraction was actually "close
to correct"? Some statistics would be helpful to understand if some more
fine-tuning of distant supervision could have helped.

Full Wikipedia results: This was the main aim of the authors and as authors
themselves said, the full system gives a performance of 26.7 (49.6 when correct
doc given, 69.5 when correct paragraph is given). Clearly, that should be a
motivation to work more on the retrieval aspect? For WebQuestions, the results
are much inferior to YodaQA, and that raises the question -- whether Wikipedia
itself is sufficient to answer all the open-domain questions? Should authors
think of an integrated model to address this? 

Overall, the final results shown in Tables 4 and 5 are inferior to some other
models. While authors only use Wikipedia, the results are not indicative of
this being the best strategy.

Other points:
The F1 value in Table 5 (78.4) is different from that in Table 4 (Both Dev and
Test).
Table 5: Why not "No f_emb"?
Error analysis: Some error analysis is required in various components of the
system. 
Are there some specific type of questions, where the system does not perform
well? Is there any way one can choose which question is a good candidate to be
answered by Wikipedia, and use this method only for those questions?
For WebQuestions, DS degrades the performance further.